# Association of albumin, fibrinogen, and modified proteins with acute coronary syndrome

**Nabila Nawar Binti, Nourin Ferdausi, Md. Eahsanul Karim Anik, Laila Noor Islam** ⓘ *

Department of Biochemistry and Molecular Biology, University of Dhaka, Dhaka, Bangladesh

* laila@du.ac.bd

## Abstract

Acute coronary syndrome (ACS) comprises a pathophysiological spectrum of cardiovascular diseases related to atherosclerotic coronary plaque erosion. Oxidative stress and inflammation play pivotal roles in the development and progression of atherosclerosis, which affects circulatory proteins, including albumin and fibrinogen, thereby causing an imbalance in albumin to globulin and fibrinogen to albumin ratios. This study aimed to assess the effect of oxidative stress on circulatory proteins, correlate these parameters, and investigate their significance in patients with ACS. In this case-control study, the major blood proteins in patients with ACS and a control group were evaluated using standard methods. Out of 70 ACS cases, 75.7% had ST-elevation myocardial infarction (STEMI), 18.6% had non-STEMI, and 5.7% had unstable angina. The mean cardiac troponin I level in patients was 12.42 ng/mL. The patients demonstrated a significantly reduced level of human serum albumin (HSA), 3.81 ± 0.99 g/dL, compared to controls, 5.33 ± 0.66 g/dL. The albumin to globulin ratio (AGR) was significantly depressed in patients while their mean fibrinogen level and the fibrinogen to albumin ratio (FAR) were significantly higher. Multivariate logistic regression analysis showed that albumin and fibrinogen were significantly associated with the risk of ACS, showing the potential of these parameters to be used for risk assessment of ACS. The ischemia modified albumin (IMA) and protein carbonyls were significantly higher in patients which showed significant positive correlations with FAR. Albumin, IMA and protein carbonyls were found to have high diagnostic sensitivity and specificity for ACS. Overall, these circulatory and modified proteins in ACS patients, particularly lower HSA, AGR, and higher IMA and protein carbonyls may help assess risk.

## Introduction

Cardiovascular diseases (CVD) are the most common causes of mortality, accounting for 31% of all deaths worldwide [1]. Coronary artery disease (CAD) is a form of CVD that contributes to the majority of cardiovascular deaths, and it is caused by the impedance of one or more arteries that deliver blood to the heart, usually due to atherosclerosis. Atherosclerotic plaque rupture or erosion, with varying degrees of superimposed thrombosis, has been demonstrated to result in a partial or complete blockage of the blood flow in the coronary arteries. The most

**Editor:** Arturo Cesaro, University of Campania Luigi Vanvitelli Department of Translational Medicine: Universita degli Studi della Campania Luigi Vanvitelli Dipartimento di Scienze Mediche Traslazionali, ITALY

**Data Availability Statement:** All relevant data are within the manuscript.

**Funding:** This study was funded by a research grant awarded to Dr. Laila Noor Islam by the Ministry of Science and Technology, Government of the People's Republic of Bangladesh. The funders had no role in study design, data collection and analysis, decision to publish, or preparation of the manuscript.

**Competing interests:** The authors declare no competing interest regarding the publication of this paper.

ominous manifestation of CAD is acute coronary syndrome (ACS), which refers to the spectrum of clinical presentations comprising unstable angina (UA), non-ST-elevation myocardial infarction (NSTEMI, partial blockage of the coronary artery), and ST-elevation myocardial infarction (STEMI, full blockage of the coronary artery) [2]. It has been reported that survivors of myocardial infarction (MI) attack are at a significant risk of recurrent MI, as well as other CVD symptoms [3].

The most common risk factor for atherosclerosis is the increased production of reactive oxygen species (ROS). Oxidative stress plays a major role in the pathophysiology of ACS [4]. Oxidative stress profoundly affects circulating proteins, and albumin is the most common among these proteins [5]. ROS produced during ischemia generate highly reactive hydroxyl free radicals in the presence of copper, causing site-specific damage to albumin at the N-terminus [6], which decreases the capacity of albumin to bind transition metals, notably, cobalt [7]. This altered form of albumin is designated ischemia modified albumin (IMA), which is a marker of oxidative stress induced protein modification with diagnostic potential in acute myocardial ischemia [8]. Another indicator of protein damage is protein carbonyls, which are produced on amino acid side chains when they are oxidized. Oxidative modifications of polypeptide chains can also contribute to the alterations of protein functions [9].

Serum albumin has essential antioxidant properties, and its lower concentration has been shown to be associated with an increased risk of cardiovascular mortality [10]. Reduced and impaired biological functions of human serum albumin (HSA) are implicated in the pathogenesis of CVD [11]. Also, low albumin to globulin ratio (AGR) has been associated with vascular adverse events and red blood cell aggregability in both acute and chronic CVD [12], including its risk factors such as old age, diabetes, hypertension, and renal insufficiency [13].

Epidemiological studies and clinical observations have found elevated levels of fibrinogen to be independently related to cardiovascular risks [14]. Fibrinogen can be involved in the early stages of atherosclerotic plaque formation. High levels of plasma fibrinogen increase the speed of platelet aggregation and the reactivity of platelets [14]. An increased level of fibrinogen has been suggested as a coronary risk indicator since it reflects the inflammatory condition of the vascular wall [15]; therefore, further understanding of the association could prevent adverse outcome of ACS. The severity of atherosclerosis can be predicted by the fibrinogen to albumin ratio (FAR). Patients with higher FAR are more likely to develop ischemia, and the FAR could be a helpful early diagnostic test for predicting ischemia [16].

Therefore, this study was conducted to investigate the levels of albumin, fibrinogen, FAR, and AGR in patients with ACS and compare the findings with a non-ACS control group. Furthermore, the serum IMA and protein carbonylation were also compared between ACS patients and controls. It was hoped that correlation among these parameters and their association with ACS would reveal some inexpensive, easy to perform, and reliable biomarkers to be used for risk assessment in ACS.

## Materials and methods

### Subjects

This case-control study was conducted on 130 adult, male participants, aged 30–70, comprising 70 patients suffering from ACS (cases), admitted to the coronary care unit of Dhaka Medical College Hospital, and 60 healthy subjects enrolled from the local community (controls). The study was approved by the institutional ethics committee that conformed to the Helsinki declaration and had been conducted at the Department of Biochemistry and Molecular Biology, University of Dhaka, Bangladesh. All subjects from both the case and the control groups gave full consent to be included in this investigation.

## Inclusion and exclusion criteria

The inclusion criteria of ACS cases were based on characteristic electrocardiogram (ECG) and troponin I value, diagnosed by expert physicians. Patients with acute chest pain and persistent (>20 min) ST-segment elevation were classified as STEMI. Patients with acute chest pain without persistent ST-segment elevation but with elevated troponin I were diagnosed with NSTEMI. Patients who presented with chest pain with no troponin I changes and with normal or undermined ECG were classified as UA. For the control group, apparently healthy volunteers who did not have CVD or any other diseases known to develop oxidative stress were enrolled. Simple random and availability sampling was applied to collect samples. Any patient or control subject suffering from any disease known to develop oxidative stress, including diabetes mellitus, infectious diseases, impaired liver, or renal functions, were excluded from the study. All subjects enrolled in the study had blood glucose levels below 6.5 mmol/L since hyperglycemia also induces oxidative stress, and is common during ACS [17], to avoid false positive results.

## Study procedures

This study was conducted from March 2018 to February 2020. In a carefully pre-designed questionnaire, all the general information for each study subject was recorded. The general information included their age, height, weight, blood pressure, family history of CVD, and hypertension. In the case of patients, additional information regarding the duration of chest pain and cardiac troponin levels were also obtained. About 10 mL of venous blood was drawn from each subject, 5 mL was collected into an EDTA containing lavender top tube for plasma collection, and the remainder was taken in a glass tube for serum collection. The serum and plasma were separated, collected in small aliquots, and stored at -20˚C until analyzed. The human serum albumin (HSA) level was estimated using bromocresol green method [18]. Briefly, 3 mL of the reagent was added to 20 μL of diluted (1:1) serum sample, mixed, and incubated for 5 minutes. HSA standards were treated in the same way. The concentration of HSA was calculated from the standard graph. The globulin contents of the samples were obtained by direct measurement of total protein (biuret method) and HSA, according to the following equation:

$$\text{Globulin concentration} = \text{Total protein concentration} - \text{HSA concentration}$$

The AGR was estimated by dividing the albumin concentration by the globulin concentration. The plasma fibrinogen level was determined using a thrombin reagent, as described previously [19]. The FAR was calculated from the values obtained by direct measurement of fibrinogen and HSA. The IMA in the serum was determined according to the method of Bar-Or et al. [7] and detailed previously [20]. Protein oxidation was estimated by measuring the protein carbonyl content of the plasma samples by the method described by Levine et al. [21] and detailed previously [22].

## Statistical analyses

GraphPad Prism (version 8.0.1, GraphPad Software, USA) was used to conduct independent samples t-test to compare the continuous variables of the two groups and graphical presentation of the analyzed data. For each parameter, the mean ± SD values were computed. The chi-squared/ Fisher's exact test was used to compare categorical variables. The Spearman correlation analysis, univariate and multivariate logistic regression analyses and the receiver operating characteristic (ROC) curve analysis were done using the Statistical Package for Social Sciences

(version 20.0, SPSS Inc., USA). The odds ratio (OR) and 95% confidence interval (CI) were reported. The results were considered significant when the value of *p* was <0.05.

## Results

### Baseline features of the study participants

Of the total 70 ACS cases, 53 (75.7%) had STEMI, 13 (18.6%) were ailed with NSTEMI, and 4 (5.7%) had UA. The duration of chest pain, from onset to hospitalization of the patients varied from 0.5 to 120 hours, with a median of 6.0 hours. The cardiac troponin I (cTnI) values, which were measured at the hospital upon diagnosis of ACS, varied from 0.01 to 78 ng/mL with a median of 1.41 ng/mL. The previous history of the patients showed 9 had heart attack or MI, 5 had angina, and 5 suffered cardiac arrest. A comparison of the baseline characteristics of the study participants is shown in Table 1. It was found that the systolic blood pressure (SBP), diastolic blood pressure (DBP), and body mass index (BMI) were not significantly different between the two groups while the mean age of the cases was significantly higher. The family history of CVD and previously diagnosed hypertension were found to be significant risk factors of ACS.

### Level of human serum albumin

The HSA concentration of the ACS patients varied from 1.70 to 6.27 g/dL with a mean of 3.81 ± 0.99 g/dL, and that in the control subjects varied from 3.81 to 7.24 g/dL with a mean of 5.33 ± 0.66 g/dL (p<0.001). For close comparison, the HSA values were further divided into three categories: lower than 4.0 g/dL, 4.0–6.0 g/dL, and greater than 6.0 g/dL. Of the patients, 57.1% had HSA lower than 4.0 g/dL, 40.0% had values between 4.0 and 6.0 g/dL, and 2.9% had HSA levels greater than 6.0 g/dL; while among the controls, 1.7% had HSA lower than 4.0 g/dL, 82.8% had values between 4.0 and 6.0 g/dL, and 15.5% had HSA levels greater than 6.0 g/dL. A chi-squared test showed significant difference between the cases and controls for the categories mentioned above (Fig 1). Among the ACS group having HSA lower than 4.0 g/dL, 42.9% were STEMI, 12.8% were NSTEMI, and 1.4% were UA patients.

### Albumin to globulin ratio

The AGR in the cases varied from 0.25 to 2.67 with a mean of 1.15 ± 0.58, and the corresponding value in the controls varied from 0.57 to 5.46 with a mean of 2.02 ± 1.18 (*p*<0.0001). Within the spectrum of ACS cases, the mean AGR in STEMI (1.11 ± 0.55) and NSTEMI

**Table 1. Baseline features of the ACS cases and the controls.**

| Variables | ACS group | Control group | p-value |
|---|---|---|---|
| BMI (kg/m$^2$) | 26.26 ± 1.41 | 26.25 ± 3.21 | 0.98 |
| SBP (mmHg) | 127.79 ± 28.32 | 122.33 ± 10.10 | 0.21 |
| DBP (mmHg) | 83.21 ± 16.83 | 81.46 ± 7.06 | 0.50 |
| Age (years) | 51.04 ± 9.83 | 44.35 ± 9.52 | <0.01 |
| DCP (hours) | 17.81 ± 24.98 | N/A | N/A |
| Troponin I (ng/mL) | 12.42 ± 19.03 | N/A | N/A |
| Hypertension (%) | 51.43 | 8.33 | <0.0001 |
| Family History of CVD (%) | 28.57 | 11.67 | <0.05 |

DCP = Duration of chest pain, N/A = Not applicable.

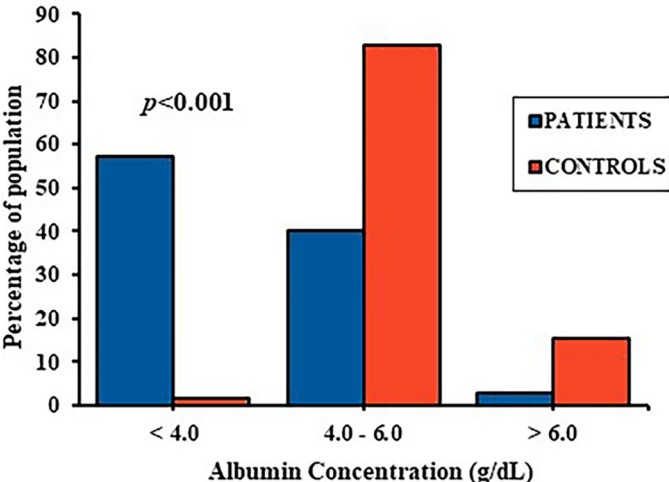

**Fig 1. Distribution of human serum albumin (HSA) levels within the ACS patient group and control group.**

(1.15 ± 0.64) cases were significantly lower than the controls, while the UA cases with a mean of 1.69 ± 0.73 were not significantly different (Fig 2a). For further analysis, the AGR values were divided into three categories: lower than 1.5, 1.5–2.0, and greater than 2.0. A significant difference between the proportions of controls and ACS patients in different categories was found in the chi-squared test (Fig 2b). Among the 71.4% of ACS cases with AGR below 1.5, 57.1% were STEMI, 12.9% were NSTEMI, and 1.4% had UA.

## Level of fibrinogen and fibrinogen to albumin ratio

The mean fibrinogen level of the controls was found to be 29.2 ± 9.2 mg/dL, ranging from 19.3–73.5 mg/dL. On the other hand, the corresponding level of the patients was 60.5 ± 57.9

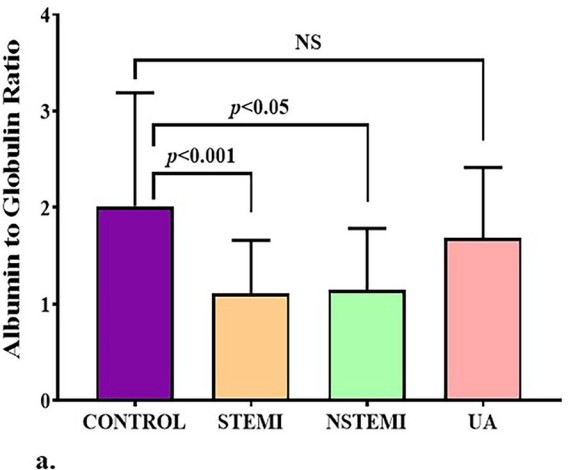
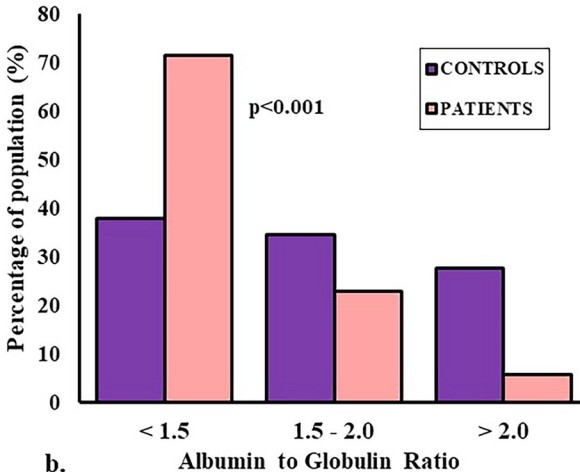

**Fig 2. (a) Comparison of albumin to globulin ratios (AGR) between the controls and different subgroups of ACS cases.** The mean AGR in STEMI and NSTEMI cases were significantly lower than the controls, while the differences in UA cases were not significant (NS). **(b) Comparison of the proportions of controls and ACS patients in different categories**. Of the ACS patients, 71.4% had AGR lower than 1.5, 22.9% with AGR between 1.5 and 2.0, and 5.7% had values greater than 2.0; among the controls, 37.9% had AGR below 1.5, 34.5% ranged between 1.5 and 2.0, and 27.6% had AGR greater than 2.0 (p<0.001, controls vs. patients, all categories).

mg/dL, with the values varying from 17.6–331.3 mg/dL. The mean fibrinogen levels in the STEMI, NSTEMI, and UA cases were 60.6, 70.8, and 27.9 mg/dL, respectively. In the control group, the mean FAR was 0.0058 ± 0.0022, whereas the value was 0.0186 ± 0.0267 in the ACS group, which was significantly higher ($p < 0.001$). Within the spectrum of ACS cases, the FAR of STEMI and NSTEMI patients varied significantly from the control group, with the mean values 0.0200 ± 0.0294 and 0.0174 ± 0.0170, respectively. The mean FAR in UA cases was 0.0053 ± 0.0007, which did not vary significantly from the controls.

## Serum IMA and protein carbonyls levels in the cases and controls

Investigation of serum IMA revealed a significant ($p < 0.0001$) elevation in the ACS group compared to the control group, and the mean values were 2.18 ± 0.61 U/mL and 1.52 ± 0.38 U/mL, respectively. The control group had a mean protein carbonyl level of 1.63 ± 1.06 nmol/mg compared to 3.16 ± 1.29 nmol/mg in the ACS group, which was significantly higher ($p < 0.0001$).

## Correlation between different parameters

A significant positive correlation was found between serum IMA and protein carbonyls in ACS patients (Fig 3a) with a Spearman correlation coefficient, ρ, of 0.317 ($p < 0.05$). There were also significant positive correlations between cTnI and fibrinogen, IMA and fibrinogen, IMA and FAR (Fig 3b), protein carbonyls and fibrinogen, and protein carbonyls and FAR (Fig 4a) in patients with ρ values of 0.508 ($p < 0.05$), 0.321 ($p < 0.05$), 0.387 ($p < 0.01$), 0.300 ($p < 0.05$), and 0.362 (p = 0.01), respectively. No such correlations were observed in the controls. Further, a significant, strong negative correlation between HSA and cTnI with a ρ of -0.643 ($p < 0.01$) was also seen in the cases (N = 30) who were brought to the hospital within 7.5 hours (median) of the onset of chest pain (Fig 4b).

## Diagnostic value of different parameters in ACS

The receiver operating characteristics (ROC) curve was analyzed to assess the clinical sensitivity and specificity of different parameters in ACS patients and controls (Fig 5a). Albumin was

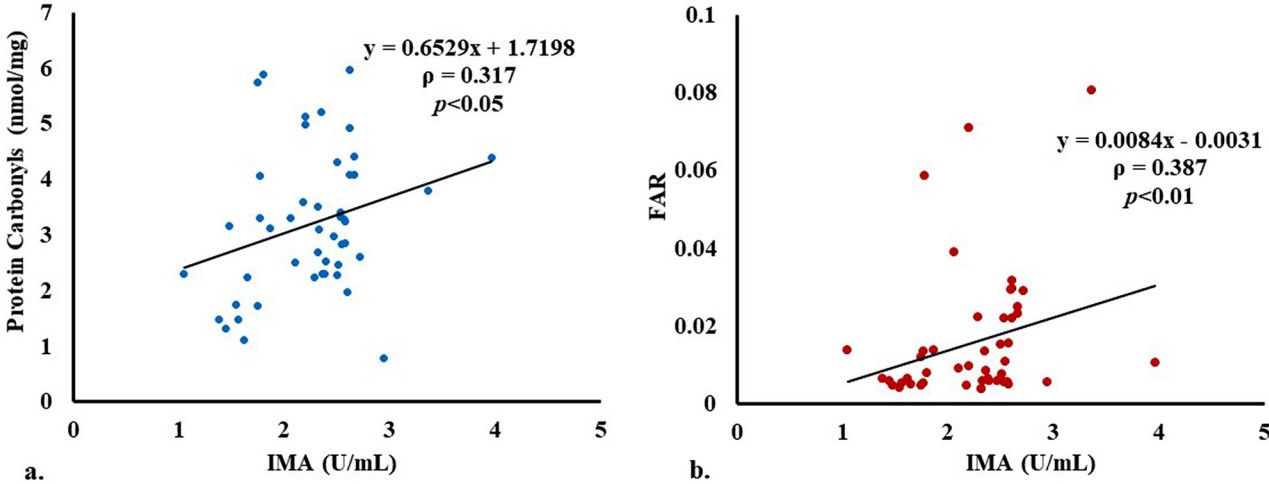

**Fig 3. (a) A significant positive correlation between IMA and protein carbonyls in ACS patients with a rho value of 0.317 ($p < 0.05$); and (b) between IMA and FAR in ACS cases with a rho value of 0.387 ($p < 0.01$).**

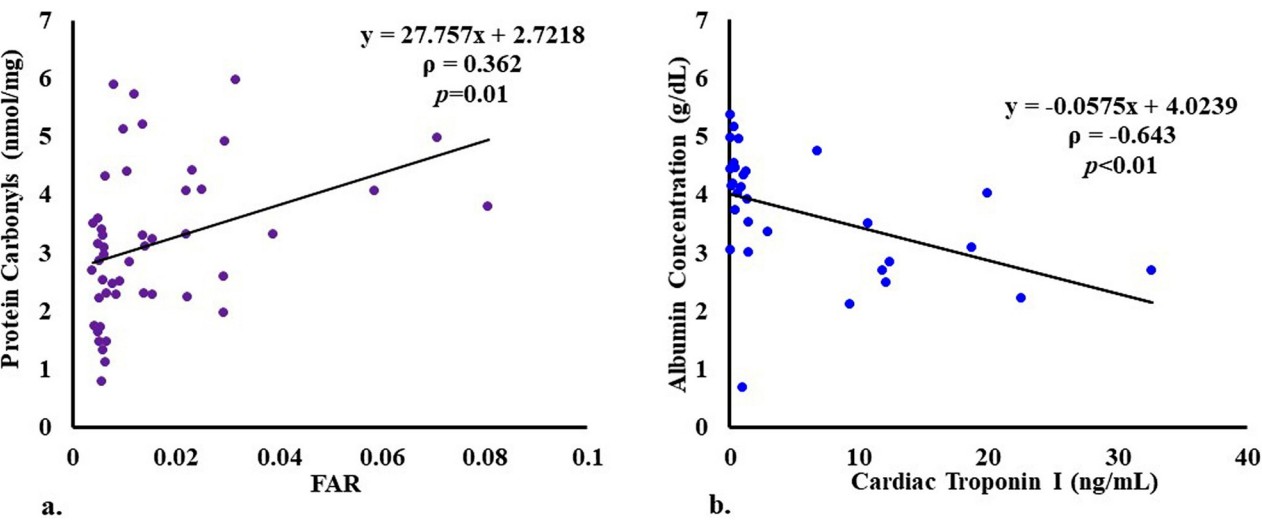

**Fig 4.** (a) A significant positive correlation between protein carbonyls and FAR in ACS patients with a rho value of 0.362 (*p* = 0.01); and (b) a significant, strong negative correlation (rho = -0.643, *p*<0.01) between serum albumin and cardiac troponin I levels in ACS cases who were brought to the hospital with acute chest pain.

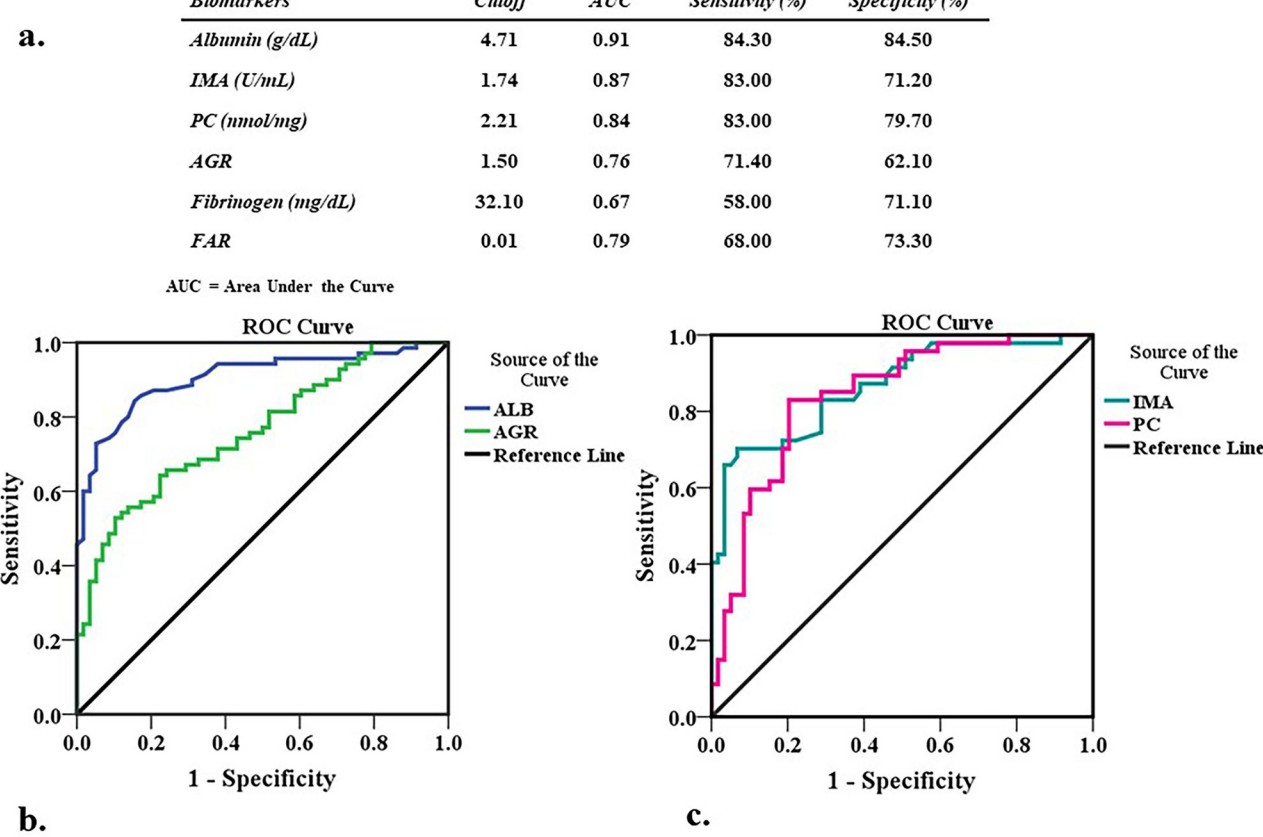

a.

| Biomarkers | Cutoff | AUC | Sensitivity (%) | Specificity (%) |
|---|---|---|---|---|
| Albumin (g/dL) | 4.71 | 0.91 | 84.30 | 84.50 |
| IMA (U/mL) | 1.74 | 0.87 | 83.00 | 71.20 |
| PC (nmol/mg) | 2.21 | 0.84 | 83.00 | 79.70 |
| AGR | 1.50 | 0.76 | 71.40 | 62.10 |
| Fibrinogen (mg/dL) | 32.10 | 0.67 | 58.00 | 71.10 |
| FAR | 0.01 | 0.79 | 68.00 | 73.30 |

AUC = Area Under the Curve

**Fig 5.** (a) The cutoff value, AUC, sensitivity and specificity of albumin, fibrinogen, AGR, FAR, IMA and protein carbonyl (PC) for distinguishing ACS from controls; (b) Receiver operating characteristic (ROC) curve of albumin (ALB) and AGR for distinguishing ACS from controls; (c) ROC curve of IMA and PC for distinguishing ACS from controls.

identified as the best biomarker in distinguishing ACS from controls, with an area under the curve (AUC) value of 0.906 (Fig 5b), followed by IMA and protein carbonyl with AUC value of 0.865 and 0.840, respectively (Fig 5c). The AUC value of AGR was 0.762, which had a cut-off value of 1.5 and provided 71.4 and 62.1% of sensitivity and specificity, respectively (Fig 5b).

## Logistic regression analyses of risk factors

To determine the potential of albumin, fibrinogen and other baseline characteristics in predicting risk of ACS, univariate and multivariate logistic regression analyses were conducted. The results (Table 2) of univariate regression analyses found albumin, fibrinogen, family history of CVD, age, hypertension all to be significant risk factors for CVD. Upon multivariate analysis, a low value of albumin level was found to be an independent risk factor for ACS (odds ratio [OR]: 0.16 [0.057–0.454], p = 0.001) after adjusting for age, hypertension, family history of CVD and fibrinogen. In addition, a high value of fibrinogen was also a powerful independent risk factor for ACS (OR: 1.042 [1.002–1.083]), $p < 0.05$) after adjusting for age, hypertension, family history of CVD and albumin (Table 2). Further, the multivariate logistic regression analysis found family history of CVD was a significant risk factor for ACS ($p < 0.05$), while age and hypertension were not.

## Discussion

This study evaluated alterations in circulatory proteins in patients with ACS. Epidemiologic studies have reported on associations of various inflammatory factors, including low albumin with coronary heart disease (CHD) [10,23,24]. Since hyperglycemia decreases regenerative potential of the myocardium [25], we excluded patients and controls with hyperglycemic random blood sugar levels from our investigation, to evaluate the association of albumin, fibrinogen and modified proteins with ACS independently from hyperglycemia.

In the present study, a significantly lower albumin concentration in the ACS patients suggested that lower albumin levels are indeed associated with ACS. In addition, this study revealed that a low value of albumin level was an independent risk factor for ACS and albumin level lower than 4.71 g/dL had 84.3% sensitivity and 84.5% specificity in identifying individuals at higher risk of ACS. This finding in ACS patients is in accordance with a previous observation that HSA lower than 4.5 g/dL is associated with an increased risk of CHD incidence [23]. One of the novel findings of this study is a significant negative correlation between serum albumin and myocardial cell damage marker, cardiac troponin I, which corroborates the association of low albumin with ACS. These findings suggest that regular monitoring and taking preventive measures to increase HSA concentrations may serve to reduce the risk of ACS.

**Table 2. Univariate and multivariate logistic regression analyses of risk factors of ACS.**

| Variable | Univariate OR (95% CI) | Univariate (p-value) | Multivariate OR (95% CI) | Multivariate (p-value) |
|---|---|---|---|---|
| Albumin | 0.097 (0.043–0.218) | <0.001 | 0.16 (0.057–0.454) | 0.001 |
| Fibrinogen | 1.047 (1.016–1.079) | 0.003 | 1.042 (1.002–1.083) | 0.038 |
| Family History of CVD | 3.029 (1.179–7.780) | 0.021 | 5.238 (1.269–21.626) | 0.022 |
| Age | 1.094 (1.051–1.139) | <0.001 | 1.042 (0.990–1.096) | 0.119 |
| Hypertension | 11.647 (4.164–32.576) | <0.001 | 2.344 (0.627–8.769) | 0.206 |

OR = Odds Ratio, CI = confidence interval.

Another novel finding of the present study is significantly lower AGR in STEMI and NSTEMI cases compared to controls, which corroborated its potential in assessing risk of ACS. The present study further found that 71.4% of patients had AGR values lower than 1.5, and STEMI and NSTEMI cases pooled together comprised 70% of the total patients with such low value of AGR. In one study, an AGR of 1.45 or less was prognostic for future vascular events [26]. Supporting this previous finding, the present study proposes an AGR lower than 1.5 to be considered high risk towards ACS occurrence, which has a sensitivity of 71.4% and a specificity of 62.1%. Therefore, this simple and inexpensive parameter may be considered for risk assessment of ACS.

Plasma fibrinogen level is an independent risk factor for CVD [27]. In this study, the fibrinogen levels in ACS cases were significantly higher than in the controls and a high value of fibrinogen was found to be an independent risk factor for ACS, which is in agreement with another study [28]. This finding indicates the potential of keeping the fibrinogen levels controlled in reducing the risk of ACS. However, it should be mentioned that the fibrinogen values in this study were lower than those reported in the study of Shi et al [30], both for the cases and controls. The present study further found a significant positive correlation of fibrinogen with cTnI. A similar correlation was shown in another study demonstrating the coexistence of inflammation and cardiac injury in patients with ACS [29]. One of the critical findings of the present study is significantly higher FAR in the patients. This observation further supports reports of increased FAR in patients with cardiovascular events, as in the literature [16], and indicates the significance of considering FAR as a predictor of future adverse outcomes of ACS.

Apart from inflammation, myocardial ischemia is the most common underlying cause of ACS. Recent research has found IMA to be an ideal biomarker for ischemia [8], which has been described as a sensitive biomarker for identifying ACS in patients presenting with acute chest pain [30] and thus, a possible diagnostic tool. In the present study, a significantly higher IMA was recorded in ACS cases, which was consistent with previous findings [20,31]. The present study found an IMA level greater than 1.74 U/mL has a diagnostic sensitivity and specificity of 83.0 and 71.2%, respectively and may be used as a reference value upon further validation. Furthermore, the positive correlations between IMA and fibrinogen, and IMA and FAR are indicative of the role of increased fibrinogen, decreased albumin, and increased FAR in ischemia.

The most widely used biomarker for oxidative damage to proteins is protein carbonyls that reflects cellular damage induced by multiple forms of ROS [9]. In the present study, the protein carbonyls of the ACS cases were significantly higher than the controls, which agreed with previous findings [22,32]. In addition, a protein carbonyl level greater than 2.21 nmol/mg with a sensitivity and specificity of 83.0 and 79.7% showed its association with ACS. Further, a significant positive correlation between protein carbonyls and IMA indicated that both these markers of oxidative protein damage are elevated in ACS patients, which may serve as diagnostic tools in the future. Moreover, significant positive correlations of fibrinogen, and FAR with protein carbonyls suggested their association with oxidative stress-mediated protein damage in patients with ACS.

It is well established that hypertension damages the arteries by making them less elastic which decreases the flow of blood and oxygen to the heart; on the other hand, age can cause the development of additional risk factors such as obesity which may also affect the heart. Although there was a significant difference in age and hypertension between cases and controls, multivariate logistic regression analysis did not find age and hypertension to be significant risk factors for ACS, after adjusting for other variables in this study. This could be because in our study population, age and hypertension were weaker risk factors compared to the other biochemical variables.

## Limitations and suggestions

There are some limitations to this investigation. The ACS cases were enrolled from only one hospital, and the study population was relatively small. Additionally, all patients admitted to the hospital with ACS during the study period could not be enrolled due to different comorbid conditions and did not take into account differences in hospital treatment procedures. Finally, the mean age of the ACS group was higher by about 6.7 years than the control group, which was due to difficulties in finding age-matched controls with the stringent inclusion criteria. However, multivariate logistic regression analysis did not find age as a significant risk factor for ACS, after adjusting for other variables including albumin, fibrinogen. Albeit these limitations, the significant findings of the study can be used as a reference for future multicenter studies with larger sample size and long-term follow-up, considering the significance of the studied parameters. Findings of this study suggest monitoring lower albumin levels to avoid adverse outcomes.

## Conclusion

The present study found significantly lower levels of albumin and AGR, and higher levels of fibrinogen in patients with ACS, which could be promising serum biomarkers for assessing risk and improving risk stratification in ACS. Increased protein carbonyls and IMA in patients suggest the presence of ongoing oxidative stress and ischemia, respectively, and correlations of these parameters with each other and independently with fibrinogen or FAR indicate the role of increased fibrinogen in ischemia and its association with protein alteration in patients with ACS. Finally, albumin, IMA and protein carbonyl levels may be exploited to detect risk of ACS, owing to their high diagnostic sensitivity and specificity.

## Acknowledgments

The authors gratefully thank Professor Dr. Abdul Wadud Chowdhury, Head, Department of Cardiology, Dhaka Medical College Hospital, for his permission and cooperation in collecting patient samples. The authors wish to thank all participants of this study.

## Author Contributions

**Conceptualization:** Laila Noor Islam.

**Formal analysis:** Nourin Ferdausi, Md. Eahsanul Karim Anik.

**Funding acquisition:** Laila Noor Islam.

**Investigation:** Nabila Nawar Binti.

**Methodology:** Nabila Nawar Binti, Nourin Ferdausi, Md. Eahsanul Karim Anik.

**Resources:** Laila Noor Islam.

**Supervision:** Laila Noor Islam.

**Visualization:** Laila Noor Islam.

**Writing – original draft:** Nabila Nawar Binti.

**Writing – review & editing:** Laila Noor Islam.

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
