## [Decision Letter · Decision Letter 0]

20 Jan 2022

PONE-D-21-36646Significance of Assessing Circulatory Proteins in Patients with Acute Coronary SyndromePLOS ONE

Dear Dr. Islam,

Thank you for submitting your manuscript to PLOS ONE. After careful consideration, we feel that it has merit but does not fully meet PLOS ONE’s publication criteria as it currently stands. Therefore, we invite you to submit a revised version of the manuscript that addresses the points raised during the review process.

We look forward to receiving your revised manuscript.

Kind regards,

Arturo Cesaro, MD

Academic Editor

PLOS ONE

Journal Requirements:

“This study was funded by a research grant awarded to Dr. LNI by the Ministry of Science and Technology, Government of the People’s Republic of Bangladesh.”

Additional Editor Comments:

Please edit the manuscript carefully as suggested by the reviewers.

Check reference style, word count by consulting the instructions for authors.

Reviewers' comments:

Reviewer's Responses to Questions

**Comments to the Author**

1. Is the manuscript technically sound, and do the data support the conclusions?

Reviewer #1: Yes

Reviewer #2: Partly

Reviewer #3: Partly

2. Has the statistical analysis been performed appropriately and rigorously? 

Reviewer #1: Yes

Reviewer #2: Yes

Reviewer #3: Yes

3. Have the authors made all data underlying the findings in their manuscript fully available?

Reviewer #1: Yes

Reviewer #2: Yes

Reviewer #3: Yes

4. Is the manuscript presented in an intelligible fashion and written in standard English?

Reviewer #1: Yes

Reviewer #2: No

Reviewer #3: Yes

5. Review Comments to the Author

Reviewer #1: The study is interesting and quite well written. The conclusions, if confirmed by larger studies, could have clinical relevance. The main limitation, as pointed out by the authors, is the low sample size.

This reviewer raises some issues that need to be addressed by the authors.

1- The authors included diabetes among the exclusion criteria. However, during ACS, hyperglycemia is also frequently found in non-diabetic individuals. In this setting, tight glycemic control favorably influences the CV outcomes of these patients (Journal of Clinical Endocrinology and Metabolism Volume 97, Issue 3, March 2012, 933-942. doi: 10.1210/jc.2011-2037 - Journal of Diabetes Research, 2018, art. no. 3106056. doi: 10.1155/2018/3106056). The text and tables are missing glucose values during hospitalization for ACS, which should have been included in the multivariate analysis. This issue, which is a limitation of the study, and the above references should be discussed in the manuscript.

2- In multivariate logistic regression analyzes, age and hypertension are not independently correlated with ACS. How do the authors justify these unexpected results?

Reviewer #2: The authors conducted “Assessing Circulatory Proteins in Patients with ACS”. While the idea is interesting, I have several concerns.

• The major weakness of this article is that extensive editing of the English language and style is required. Indeed, some parts are absolutely not comprehensible.

• The article’s main title is ambiguous and should be rephrased to be consistent with the precise goal.

• Line 55-58: Please indicate the source of this text.

• Line 59-61: This text does not appear to be in keeping with the preceding text's context, please find sources that narrate the dangerous events of this disease

• Please pay attention to writing the main and sub-headings of the article, and follow the basic approaches to writing scientific research

• Line 100: Please change the title “Subjects and methods” to “Materials and Methods”

• How was the sample size calculated, and is the number of specific subjects sufficient for this study, please explain it in detail

• Why did you specify the ages of the participants from 30 to 70 years? If you mean that the participants were within the ages you specified, then you must transfer this information to the results section and include the acceptable ages in the study.

• The inclusion and exclusion criteria are confusing and unclear.

• Please add a paragraph entitled Study Procedures, and explain all the details of the study to be clear to the reader

• Please delete the "Study period and blood sample collection" paragraph and move the information about it to the study procedures paragraph

• Please specify the end points of the study in a separate paragraph

• Put all the basic tests under the main title "Measurements"

• In table 1: change “ACS Cases” to “ACS group”, “Controls” to “Control group”, and “Statistics” to “p-value. Also, please find the P-values for BMI, SBP, and DBP and add them to the table.

• Line 167: Cheng the title “Level of serum albumin in the study subjects” to “level of human serum albumin”, and please illustrate the results of this test with a graph.

• finally, this study has several limitations that may affect the results of the study.

Reviewer #3: Nabila Nawar Binti, et al. demonstrate that Albumin, ischemia modified albumin (IMA) and protein carbonyls were found to have high diagnostic sensitivity and specificity for ACS. Of interest, these circulatory and modified proteins in ACS patients, particularly lower HSA, AGR, and higher IMA and protein carbonyls showed the potentials to be used for risk assessment of ACS.

The study is of interest nevertheless, I have several concerns that need to be addressed before the study could be re-submitted.

1) Introduction: please, shorten the introduction focusing on the biomarkers.

2) Methods: Please merge the paragraphs from 2.4 to 2.10 in one single session about the Labo test of the biomarkers/circulating proteins.

3) Methods: please describe how family history of CVD was assessed considering that is one of the independent factor of the multivariate logistic regression.

4) Results: “The duration of chest pain, from onset to hospitalization of the patients varied 158 from 0.5 to 120 hours, with a median of 6.0 hours”. Please, split the symptoms to balloon time for STEMI and NSTEMI/UA.

5) Results: dyslipidemia, previous history of CAD and admission medical therapy should be added to table 1 in order to better describe the 2 study populations.

6) Figure Legends: I am not sure that the figure legend should be in the text. Please, verify.

7) Results: Please merge the paragraphs 3.4 - 3.5 and 3.6 – 3.7 in two different paragraphs.

8) Results: Please add some information regarding the angiographic data (vessels affected by the lesions, PCI performed, numbers of stent).

9) Results: were there any patients without significant coronary artery disease (MINOCA)? Please clarify this information.

10) Results: Please, can the authors provide some data about the standard inflammatory agents such as WBC (white blood cells), neutrophils and lymphocytes counts and CRP.

11) Results: Please, can the authors provide some data about the admission blood glucose level and possible correlations with these circulating inflammatory proteins (albumin, ischemia modified albumin (IMA) and protein carbonyls)?

12) Discussion: Please shorten the discussion, focusing on the main findings.

13) Discussion: Please integrate the discussion with the following ref PMID: 33530978 regarding the inflammatory burden in patients with acute myocardial infarction.

6. PLOS authors have the option to publish the peer review history of their article (what does this mean?). If published, this will include your full peer review and any attached files.

Reviewer #1: No

Reviewer #2: No

Reviewer #3: No

---

## [Author Response · Author response to Decision Letter 0]

3 Mar 2022

Journal Requirements:

Response: We ensure that our manuscript meets PLOS ONE’s style requirements, including those for file naming.

“This study was funded by a research grant awarded to Dr. LNI by the Ministry of Science and Technology, Government of the People’s Republic of Bangladesh.”

Response: In the cover letter, the following statement has been included "The funders had no role in study design, data collection and analysis, decision to publish, or preparation of the manuscript."

Additional Editor Comments:

Please edit the manuscript carefully as suggested by the reviewers.

Check reference style, word count by consulting the instructions for authors.

Response: We have edited the manuscript carefully as suggested by the reviewers. We have checked the reference style, preparation of figures, figure legends and their placement in the text, and word count by following the instructions for authors. The total number of words in our revised manuscript is 4879.

Reviewers' comments:

Reviewer's Responses to Questions

Comments to the Author

1. Is the manuscript technically sound, and do the data support the conclusions?

Reviewer #1: Yes

Reviewer #2: Partly

Reviewer #3: Partly

2. Has the statistical analysis been performed appropriately and rigorously?

Reviewer #1: Yes

Reviewer #2: Yes

Reviewer #3: Yes

3. Have the authors made all data underlying the findings in their manuscript fully available?

Reviewer #1: Yes

Reviewer #2: Yes

Reviewer #3: Yes

4. Is the manuscript presented in an intelligible fashion and written in standard English?

Reviewer #1: Yes

Reviewer #2: No

Reviewer #3: Yes

5. Review Comments to the Author

Reviewer #1: The study is interesting and quite well written. The conclusions, if confirmed by larger studies, could have clinical relevance. The main limitation, as pointed out by the authors, is the low sample size.

This reviewer raises some issues that need to be addressed by the authors.

1- The authors included diabetes among the exclusion criteria. However, during ACS, hyperglycemia is also frequently found in non-diabetic individuals. In this setting, tight glycemic control favorably influences the CV outcomes of these patients (Journal of Clinical Endocrinology and Metabolism Volume 97, Issue 3, March 2012, 933-942. doi: 10.1210/jc.2011-2037 - Journal of Diabetes Research, 2018, art. no. 3106056. doi: 10.1155/2018/3106056). The text and tables are missing glucose values during hospitalization for ACS, which should have been included in the multivariate analysis. This issue, which is a limitation of the study, and the above references should be discussed in the manuscript.

Reply to the reviewer: As mentioned by the learned reviewer, the stated references observed that tight glycemic control may increase regenerative potential of the ischemic myocardium (J Clin Endocrinol Metab. 2012; 97:933-942, doi: 10.1210/jc.2011-2037), and hyperglycemia is common during ACS which is a significant and independent mortality predictor among diabetic patients with recent ACS (J Diabetes Res. 2018, art. no. 3106056. doi: 10.1155/2018/3106056). Considering these, we excluded patients and controls with hyperglycemic random blood sugar levels from our study, so that the association of albumin, fibrinogen and modified proteins with ACS could be assessed independently from hyperglycemia. It should be mentioned here that all patients enrolled in the study had blood glucose levels below 6.5 mmol/L on admission. 

2- In multivariate logistic regression analyzes, age and hypertension are not independently correlated with ACS. How do the authors justify these unexpected results?

Reply to the reviewer: To determine the potential of albumin, fibrinogen and other baseline characteristics in predicting risk of ACS, both univariate (not shown in the manuscript) and multivariate logistic regression analyses were conducted. In univariate logistic regression, age, hypertension, family history of CVD, albumin and fibrinogen all were independent risk factors for ACS. Upon multivariate logistic regression analysis, age and hypertension were found to be not significant with respect to all other parameters.

Reviewer #2: The authors conducted “Assessing Circulatory Proteins in Patients with ACS”. While the idea is interesting, I have several concerns.

• The major weakness of this article is that extensive editing of the English language and style is required. Indeed, some parts are absolutely not comprehensible. 

Reply to the reviewer: We have edited the Introduction, Materials and Methods, and Discussion sections to make them more comprehensible. We believe, the learned reviewer would find an overall improvement of language in the Revised Manuscript.

• The article’s main title is ambiguous and should be rephrased to be consistent with the precise goal.

Reply to the reviewer: The title of the article has been rephrased to be consistent with the precise goal of the study:

Association of albumin, fibrinogen, and modified proteins with acute coronary syndrome 

 • Line 55-58: Please indicate the source of this text.

Reply to the reviewer: A reference has been added in the manuscript to indicate the source of the text.

• Line 59-61: This text does not appear to be in keeping with the preceding text's context, please find sources that narrate the dangerous events of this disease

Reply to the reviewer: This has been addressed in the manuscript.

• Please pay attention to writing the main and sub-headings of the article, and follow the basic approaches to writing scientific research

Reply to the reviewer: The main and sub-headings of the article have been changed and improved.

• Line 100: Please change the title “Subjects and methods” to “Materials and Methods”

Reply to the reviewer: The title has been changed in the manuscript.

• How was the sample size calculated, and is the number of specific subjects sufficient for this study, please explain it in detail.

Reply to the reviewer: Acute coronary syndrome (ACS, formerly called ischemic heart disease) is caused by a sudden onset of cardiac tissue ischemia secondary to impaired blood flow. Bangladeshi people have high susceptibility to ischemic heart disease (IHD) although no population-based data is available. One study found the prevalence of IHD in Bangladeshi men to be 4.6% (Zaman et al, 2007; PMID: 19124932). It may be mentioned here that all our study participants were men, which had been inadvertently omitted from the manuscript. It has now been mentioned under the heading “Subjects” in the manuscript. Therefore considering the available prevalence rate, and owing to the difficulties in collecting blood samples from patients admitted to the CCU by following the stringent inclusion criteria of this study, we had to consider a single parameter (namely, human serum albumin) to calculate the sample size. The formula used is:

n = Z2p(1-p)/d2, where n = sample size; Z = confidence level at 95%; p = expected prevalence (here, 4.6%); and d = precision (±5%). 

By substituting all values in the formula, we get:

n = 1.962×0.046(1-0.046)/0.052 = 67.4

Thus, we collected 70 samples from ACS patients. 

Please note that this number is not sufficient as there are other parameters in this study. We have mentioned this in the limitations of our study. 

• Why did you specify the ages of the participants from 30 to 70 years? If you mean that the participants were within the ages you specified, then you must transfer this information to the results section and include the acceptable ages in the study. 

Reply to the reviewer: In a recent study carried out in Bangladesh on 800 hospitalized patients with ACS, it was found that approximately 5% of patients were less than 30 years of age while about 10% were aged 70 years or more (Ahmed et al., 2018; doi.org/:10.3329/bhj.v33i1.37018). We found a similar scenario in our preliminary study; hence we selected this age range to enroll study participants.

• The inclusion and exclusion criteria are confusing and unclear.

Reply to the reviewer: We have revised the relevant section and clarified the inclusion and exclusion criteria of the study subjects in the manuscript.

• Please add a paragraph entitled Study Procedures, and explain all the details of the study to be clear to the reader

Reply to the reviewer: We have added a paragraph entitled Study Procedures, and explained all the details of the study to be clear to the reader.

• Please delete the "Study period and blood sample collection" paragraph and move the information about it to the study procedures paragraph

Reply to the reviewer: We have provided the information on study period and blood sample collection to the study procedures paragraph.

• Please specify the end points of the study in a separate paragraph

Reply to the reviewer: Since our study was a case-control study, we did not specify any end points. In future similar observational cohort studies or clinical trials, the study end points could include all cause mortality, recurrent MI, cardiovascular deaths, stroke, etc.

• Put all the basic tests under the main title "Measurements"

Reply to the reviewer: All the basic tests have been put under the heading “Study procedures”, as suggested by Reviewer #3 (point no. 2, below).

• In table 1: change “ACS Cases” to “ACS group”, “Controls” to “Control group”, and “Statistics” to “p-value. Also, please find the P-values for BMI, SBP, and DBP and add them to the table.

Reply to the reviewer: All suggestions of the learned reviewer have been incorporated in Table 1 of the revised manuscript.

• Line 167: Cheng the title “Level of serum albumin in the study subjects” to “level of human serum albumin”, and please illustrate the results of this test with a graph.

Reply to the reviewer: The change has been made in the manuscript. The results of human serum albumin in the ACS and Control groups have been illustrated with a figure (Fig 1).

• finally, this study has several limitations that may affect the results of the study.

Reply to the reviewer: The limitations of this study have been discussed in the manuscript. However, the conclusions, if confirmed by larger studies, could have clinical relevance.

Reviewer #3: Nabila Nawar Binti, et al. demonstrate that Albumin, ischemia modified albumin (IMA) and protein carbonyls were found to have high diagnostic sensitivity and specificity for ACS. Of interest, these circulatory and modified proteins in ACS patients, particularly lower HSA, AGR, and higher IMA and protein carbonyls showed the potentials to be used for risk assessment of ACS.

The study is of interest nevertheless, I have several concerns that need to be addressed before the study could be re-submitted.

1) Introduction: please, shorten the introduction focusing on the biomarkers.

Reply to the reviewer: The introduction has been shortened and focused on the biomarkers.

2) Methods: Please merge the paragraphs from 2.4 to 2.10 in one single session about the Labo test of the biomarkers/circulating proteins.

Reply to the reviewer: The paragraphs 2.4 to 2.10 have been merged in one single session under the heading “Study procedures”.

3) Methods: please describe how family history of CVD was assessed considering that is one of the independent factors of the multivariate logistic regression.

Reply to the reviewer: Details have been added in the manuscript under the heading “Study Procedures”. Briefly - in a carefully pre-designed questionnaire, all the general information for each study subject was recorded which included their age, height, weight, blood pressure, any family history of CVD, and hypertension.

4) Results: “The duration of chest pain, from onset to hospitalization of the patients varied 158 from 0.5 to 120 hours, with a median of 6.0 hours”. Please, split the symptoms to balloon time for STEMI and NSTEMI/UA.

Reply to the reviewer: Unfortunately, this information was not available from the hospital.

5) Results: dyslipidemia, previous history of CAD and admission medical therapy should be added to table 1 in order to better describe the 2 study populations.

Reply to the reviewer: The lipid profile of the study participants was measured in the study but not included in the manuscript. There were no significant differences in serum levels of cholesterol, triglycerides, and LDL cholesterol among the two study populations (it should be mentioned here that the patients were already hospitalized and under cholesterol-lowering drugs). However, the HDL cholesterol was found significantly diminished in the patient group in contrast to that of the control group. The previous history of the patients showed 9 had MI, 5 had angina, and 5 suffered cardiac arrest (this information has been added in the manuscript under the heading “Baseline features of the study participants”). The admission medical therapy data of the patients was unavailable.

6) Figure Legends: I am not sure that the figure legend should be in the text. Please, verify.

Reply to the reviewer: The PLOS ONE style template has been checked regarding the placement of the figure legend.

7) Results: Please merge the paragraphs 3.4 - 3.5 and 3.6 – 3.7 in two different paragraphs.

Reply to the reviewer: This has been addressed in the manuscript.

8) Results: Please add some information regarding the angiographic data (vessels affected by the lesions, PCI performed, numbers of stent).

Reply to the reviewer: Unfortunately, the suggested information was not available from the hospital.

9) Results: were there any patients without significant coronary artery disease (MINOCA)? Please clarify this information.

Reply to the reviewer: Clinicians did not report any such observation.

10) Results: Please, can the authors provide some data about the standard inflammatory agents such as WBC (white blood cells), neutrophils and lymphocytes counts and CRP.

Reply to the reviewer: We have done the blood platelet count (data not shown) and measured fibrinogen levels in plasma of the study participants and found both the inflammatory parameters to be significantly higher in the ACS group. However, in a recently published paper from our lab, the WBC count of ACS patients was found to be 11.76 ± 2.49 × million cells/mL compared to 7.37 ± 1.77 × million cells/mL in the control subjects (p<0.001); the neutrophil count was significantly higher and lymphocyte count was significantly lower in patients (Afr.J.Bio.Sc. 4(1) (2022) 37-47). In the present study, some of the patients had CRP level between 6.0 and 20.0 mg/L.

11) Results: Please, can the authors provide some data about the admission blood glucose level and possible correlations with these circulating inflammatory proteins (albumin, ischemia modified albumin (IMA) and protein carbonyls)?

Reply to the reviewer: All patients enrolled in the study had blood glucose levels below 6.5 mmol/L on admission. Individual values were not recorded for further analysis. 

12) Discussion: Please shorten the discussion, focusing on the main findings.

Reply to the reviewer: We have shortened the discussion, focusing on the main findings.

13) Discussion: Please integrate the discussion with the following ref PMID: 33530978 regarding the inflammatory burden in patients with acute myocardial infarction.

Reply to the reviewer: Unfortunately, we couldn’t integrate the discussion with the suggested reference (PMID 33530978) since the study focused on the interplay between hyperglycemia, inflammation and infarct size in a cohort of patients admitted with acute myocardial infarction, including cases of MINOCA, which did not exactly relate to our study.

6. PLOS authors have the option to publish the peer review history of their article (what does this mean?). If published, this will include your full peer review and any attached files.

Do you want your identity to be public for this peer review? For information about this choice, including consent withdrawal, please see our Privacy Policy.

Reviewer #1: No

Reviewer #2: No

Reviewer #3: No

Response: We uploaded our figure files (a total of 5 figures) to the PACE digital diagnostic tool to check whether the figures met PLOS requirements; all figures were converted to the accepted formats of PLOS ONE.

---

## [Decision Letter · Decision Letter 1]

20 Apr 2022

PONE-D-21-36646R1Association of albumin, fibrinogen, and modified proteins with acute coronary syndromePLOS ONE

Dear Dr. Islam,

Thank you for submitting your manuscript to PLOS ONE. After careful consideration, we feel that it has merit but does not fully meet PLOS ONE’s publication criteria as it currently stands. Therefore, we invite you to submit a revised version of the manuscript that addresses the points raised during the review process.

We look forward to receiving your revised manuscript.

Kind regards,

Arturo Cesaro, MD

Academic Editor

PLOS ONE

Journal Requirements:

Additional Editor Comments (if provided):

You are invited to consider the reviewers' comments, reported at the end of this letter, and to revise your manuscript accordingly. In the letter accompanying your resubmission, please explain your response to each of the comments. Please observe the word count and citation style. For further details, please consult the Instructions for Authors on the website

Reviewers' comments:

Reviewer's Responses to Questions

**Comments to the Author**

1. If the authors have adequately addressed your comments raised in a previous round of review and you feel that this manuscript is now acceptable for publication, you may indicate that here to bypass the “Comments to the Author” section, enter your conflict of interest statement in the “Confidential to Editor” section, and submit your "Accept" recommendation.

Reviewer #1: (No Response)

Reviewer #2: All comments have been addressed

Reviewer #3: All comments have been addressed

2. Is the manuscript technically sound, and do the data support the conclusions?

Reviewer #1: Yes

Reviewer #2: Yes

Reviewer #3: Partly

3. Has the statistical analysis been performed appropriately and rigorously? 

Reviewer #1: Yes

Reviewer #2: Yes

Reviewer #3: Yes

4. Have the authors made all data underlying the findings in their manuscript fully available?

Reviewer #1: Yes

Reviewer #2: Yes

Reviewer #3: Yes

5. Is the manuscript presented in an intelligible fashion and written in standard English?

Reviewer #1: Yes

Reviewer #2: Yes

Reviewer #3: Yes

6. Review Comments to the Author

Reviewer #1: The authors did not address the two issues I raised.

Specifically, regarding the first question, they made a comment in their answer but did not include that comment or the references suggested in the manuscript.

As for the second question, they did not address it either in the answer or in the manuscript.

Therefore, as already suggested in my first review, from my point of view the authors need to address my previous issues in their paper.

Reviewer #2: The authors adequately reply to all previous comments, and I am happy with the revised version, the manuscript is significantly improved

Reviewer #3: All comments have been addressed

7. PLOS authors have the option to publish the peer review history of their article (what does this mean?). If published, this will include your full peer review and any attached files.

Reviewer #1: No

Reviewer #2: **Yes: **Mohammed Ahmed Akkaif

Reviewer #3: No

---

## [Author Response · Author response to Decision Letter 1]

1 Jun 2022

In response to the first question raised by Reviewer #1, the following lines have been inserted in the manuscript:

Line # 109: “All subjects enrolled in the study had blood glucose levels below 6.5 mmol/L since hyperglycemia also induces oxidative stress, and is common during ACS [17], to avoid false positive results.”

Line # 245: “Since hyperglycemia decreases regenerative potential of the myocardium [25], we excluded patients and controls with hyperglycemic random blood sugar levels from our investigation, to evaluate the association of albumin, fibrinogen and modified proteins with ACS independently from hyperglycemia.”

As for the second question, we are apologetic for failing to clarify the second issue both in our manuscript and ‘Response to Reviewers’. To address this important issue, we have included an extra column in Table 2 in the accompanying revised version [PONE-D-21-36646R1], to present the statistical findings of the univariate logistic regression analyses along with those of the multivariate regression data. Also, the following lines have been inserted in the manuscript:

Line # 297: “It is well established that hypertension damages the arteries by making them less elastic which decreases the flow of blood and oxygen to the heart; on the other hand, age can cause the development of additional risk factors such as obesity which may also affect the heart. Although there was a significant difference in age and hypertension between cases and controls, multivariate logistic regression analysis did not find age and hypertension to be significant risk factors for ACS after adjusting for other variables in this study. This could be because in our study population, age and hypertension were weaker risk factors compared to the other biochemical variables.”

To our understanding, since multivariate regression analysis considers more than one factor of independent variables that influence the variability of dependent variables, the conclusion drawn is more accurate, more realistic, and nearer to the real-life situation for our study population. 

In response to the Editor comments, we have reviewed our reference list to ensure that it is complete and correct. Reference numbers 17 and 25 have been newly inserted in the Manuscript, as suggested by Reviewer #1. We have addressed the reviewers' comments in detail, and revised our manuscript accordingly.

---

## [Decision Letter · Decision Letter 2]

11 Jul 2022

Association of albumin, fibrinogen, and modified proteins with acute coronary syndrome

PONE-D-21-36646R2

Dear Dr. Islam,

We’re pleased to inform you that your manuscript has been judged scientifically suitable for publication and will be formally accepted for publication once it meets all outstanding technical requirements.

Kind regards,

Arturo Cesaro, MD

Academic Editor

PLOS ONE

Additional Editor Comments (optional):

The paper appears to be improved after changes were made based on the reviewers' comments.

Reviewers' comments:

Reviewer's Responses to Questions

**Comments to the Author**

1. If the authors have adequately addressed your comments raised in a previous round of review and you feel that this manuscript is now acceptable for publication, you may indicate that here to bypass the “Comments to the Author” section, enter your conflict of interest statement in the “Confidential to Editor” section, and submit your "Accept" recommendation.

Reviewer #1: All comments have been addressed

2. Is the manuscript technically sound, and do the data support the conclusions?

Reviewer #1: Yes

3. Has the statistical analysis been performed appropriately and rigorously? 

Reviewer #1: Yes

4. Have the authors made all data underlying the findings in their manuscript fully available?

Reviewer #1: Yes

5. Is the manuscript presented in an intelligible fashion and written in standard English?

Reviewer #1: Yes

6. Review Comments to the Author

Reviewer #1: All issues raised by this reviewer were addressed by authors. The manuscript is methodologically correct. The conclusions were supported by results. In this revised version the authors improved the original manuscript.

7. PLOS authors have the option to publish the peer review history of their article (what does this mean?). If published, this will include your full peer review and any attached files.

Reviewer #1: No

---

## [Editor Report · Acceptance letter]

18 Jul 2022

PONE-D-21-36646R2 

Association of albumin, fibrinogen, and modified proteins with acute coronary syndrome 

Dear Dr. Islam:

I'm pleased to inform you that your manuscript has been deemed suitable for publication in PLOS ONE. Congratulations! Your manuscript is now with our production department. 

Kind regards, 

on behalf of

Dr. Arturo Cesaro 

Academic Editor

PLOS ONE